# DragEntity:Trajectory Guided Video Generation using Entity and Positional Relationships

## ABSTRACT

In recent years, diffusion models have achieved tremendous success in the field of video generation, with controllable video generation receiving significant attention. However, existing control methods still face two limitations: Firstly, control conditions (such as depth maps, 3D Mesh) are difficult for ordinary users to obtain directly. Secondly, it's challenging to drive multiple objects through complex motions with multiple trajectories simultaneously. In this paper, we introduce DragEntity, a video generation model that utilizes entity representation for controlling the motion of multiple objects. Compared to previous methods, MotionCtrl offers two main advantages: 1) Trajectory-based methods are more user-friendly for interaction. Users only need to draw trajectories during the interaction to generate videos. 2) We use entity representation to represent any object in the image, and multiple objects can maintain relative spatial relationships. Therefore, we allow multiple trajectories to control multiple objects in the image with different levels of complexity simultaneously. Our experiments validate the effectiveness of DragEntity, demonstrating its superior performance in fine-grained control in video generation.

## CCS CONCEPTS

• **Computing methodologies** → *Reconstruction*.

## KEYWORDS

Trajectory , Controllable Video Generation, Entity, Positional Relationships

## 1 INTRODUCTION

Video generation, such as text-to-video (T2V) generation [5, 7, 14], aims to produce diverse and high-quality videos based on given text prompts. Unlike image generation [23, 25], which focuses on generating a single image, video generation requires creating consistent and smooth motion within the generated sequence of images. Therefore, motion control plays a crucial and significant role in video generation and has received great attention in the research of various control mechanisms in recent years.

Currently, in the field of controllable video generation, previous works mainly emphasized image-to-video generation, using an initial frame image as the control condition to generate video sequences, as seen in the works of [9, 19, 28]. However, relying

*ACM MM, 2024, Melbourne, Australia*

© 2024 Copyright held by the owner/author(s). Publication rights licensed to ACM.
ACM ISBN 978-x-xxxx-xxxx-x/YY/MM
https://doi.org/10.1145/nnnnnnn.nnnnnnn

solely on images as the control condition cannot determine the content of subsequent video frames, making it difficult to generate longer videos. As a result, research has shifted towards the text-to-video domain, using long texts or prompt words to constrain the semantic content of video generation, as in the works of [16, 27]. Recently, some studies have utilized both text and images as control conditions for more accurate control, as demonstrated by [10, 42]. Despite this, due to the ambiguity and subjectivity of language, images can only serve as initial frames for content guidance, thus the information as control conditions remains limited in aspects such as camera movement and complex object trajectories.

In the field of video generation, trajectory-based control has emerged as a user-friendly method, attracting increasing attention from researchers. CVG [13] and C2M [1] encoded images and trajectories, predicted optical flow maps and warping features as intermediate results for controllable video generation. However, warping operations often lead to unnatural distortions. To address this issue, II2VB [4] and iPOKE [3] compressed videos into a dense latent space and learned to manipulate these latent variables using RNNs. Similarly, MCDiff [34] predicted future frames in an autoregressive manner through diffusion delay. While MCDiff has shown promising results, it relies on HRNet by [33] to extract 17 key points for each individual, it can only control movements from humans, and the generated videos do not move along the prescribed trajectory paths. Moreover, MCDiff overlooks the generation of open-domain videos, significantly limiting its practical application value.

As one of the representative works, DragNUWA[41] encodes sparse trajectories into dense flow space sequences, then uses them as supervisory signals to control object motion. Similarly, MotionCtrl [36]directly encodes the trajectory coordinates of each object into a vector field, using this vector field as the condition to control object motion. These works have made significant contributions to trajectory-based controllable video generation. However, can a point on the image truly represent the object?

The aforementioned research still has some issues. Firstly, current works' consideration of trajectory control is not comprehensive enough. It is evident that a single pixel cannot represent an object, and dragging a single pixel cannot accurately control the corresponding object. Secondly, the objects within the pictures are quite small. Dealing with human datasets like TED-talks, Human3.6M, etc., multiple complex trajectories would only cause pixels close to the trajectory to move, thus leading to severe distortion of the human body. In fact, addressing this issue requires clarifying two concepts: 1) What is the entity. Identifying the specific area or entity to be dragged. 2) How to drag. How to achieve accurate dragging of the selected area, which means separating the background to be dragged from the foreground. For the first challenge, interactive segmentation[18, 35] is an effective solution. For example, using SAM [18] in the initial frame allows us to conveniently select the

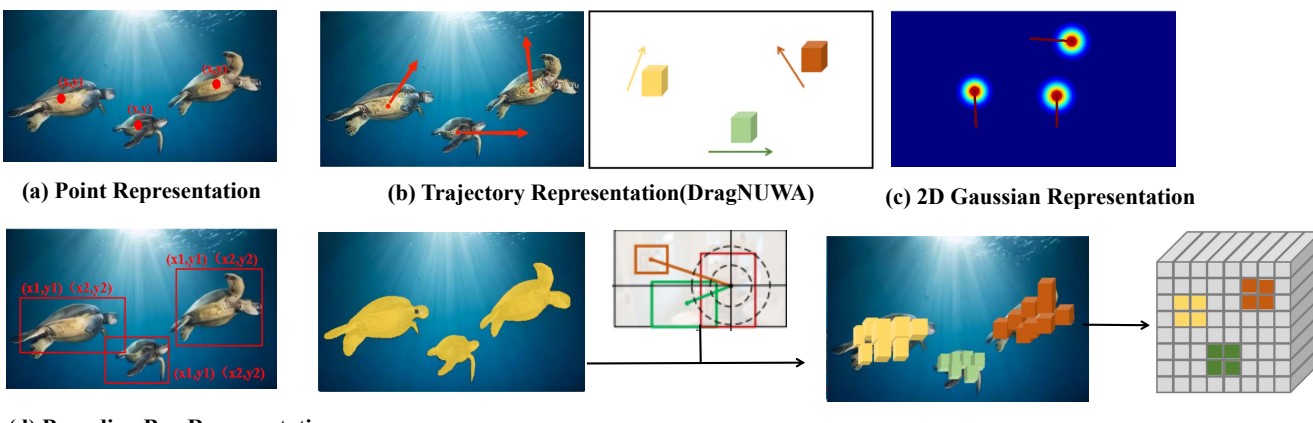

**Figure 1: Comparison of different representation modeling methods: (a) Point Representation: Represents an entity using coordinate points (x, y). (b) Trajectory Graph: Represents the trajectory of an entity using a trajectory vector graph. (c) 2D Gaussian Distribution: Represents an entity using a two-dimensional Gaussian mapping. (d) Box Representation: Represents an entity using a bounding box. (e) Entity Representation: Represents an entity using latent features that include spatial relationships between objects.**

area we wish to control. For individual human twisting, fine-grained segmentation is required, such as limbs and head. In contrast, the second technical challenge poses a greater difficulty. To address this issue, this paper proposes a novel entity representation method that integrates the spatial relationship between objects to achieve precise motion control of entities in videos and generate action videos.

Some works [8, 12, 29] have already demonstrated the effectiveness of using latent features to represent corresponding objects. Anydoor [8] utilizes the capabilities of Dino v2 [22] for object customization, while Video Swap [12]and DIFT [29] use the capabilities of diffusion models [23]for video editing tasks. DragAnything [39] uses diffusion models for instance segmentation to handle video generation tasks. Inspired by these works, we propose DragEntity, which leverages the latent features to represent individual entities. For the human body, this is further refined to a representation of 12 parts encompassing the torso and limbs. Additionally, we believe that the relative spatial relationship between entities is crucial for modeling the motion of objects. For instance, animals do not run into houses, and a human's arms cannot be below their legs.

In our work, we utilize SVD[2] as the base segmentation model (for human bodies, LIP[11] is used for fine-grained segmentation). The training requires video data, motion trajectory points, and the entity mask of the first frame. Using the mask of each entity in the first frame, we extract the center coordinates of that entity. Then, we use CoTrack[17] to predict the motion trajectory of these points as the motion trajectory of the entity.

Our main contributions are summarized as follows:

- Unlike the paradigm of dragging pixels, we propose a method for dragging objects that enables true entity-level motion control and representation, ensuring the structural integrity of the object during the dragging process.

- We introduce modeling of the relative spatial positions between objects to prevent the generation of highly unrealistic motion videos caused by trajectory dragging.

- We have conducted experiments to validate the effectiveness of DragEntity, demonstrating its superior performance in fine-grained control over video synthesis.

## 2 RELATED WORK

### 2.1 Image and Text Guided Video Generation

Text-to-video generation has been widely studied in recent years [15, 16, 27, 37, 38] introducing text descriptions to semantically control the content of video generation. However, text alone cannot accurately describe the spatial information of visuals. Therefore, MAGE Hu et al. (2022) [38]emphasizes text-image-to-video, utilizing both semantic information from text and spatial information from images for precise video control. Similarly, GEN-1 Esser et al. (2023) [23] integrates depth maps with texts using cross-attention mechanisms for control. In the domain of long video generation, text-image-to-video has also been widely used. For example, Phenaki Villegas et al. (2022)[32] generates subsequent frames by auto-regressively introducing previous frames and text, achieving long video generation. NUWA-XL Yin et al. (2023)[42] employs a hierarchical diffusion architecture to continuously complete intermediate frames based on previous frames and text. I2vgen-xl[44] introduces a cascaded network that improves model performance by separating these two factors and ensures data alignment by incorporating static images as essential guidance. Apart from academic research, the industry has also produced numerous notable works, including Gen-2 [13][10], and SORA[6]. However, compared to the general video generation efforts, the development of controllable video generation still has room for improvement. In our work, we aim to advance the field of trajectory-based video generation.

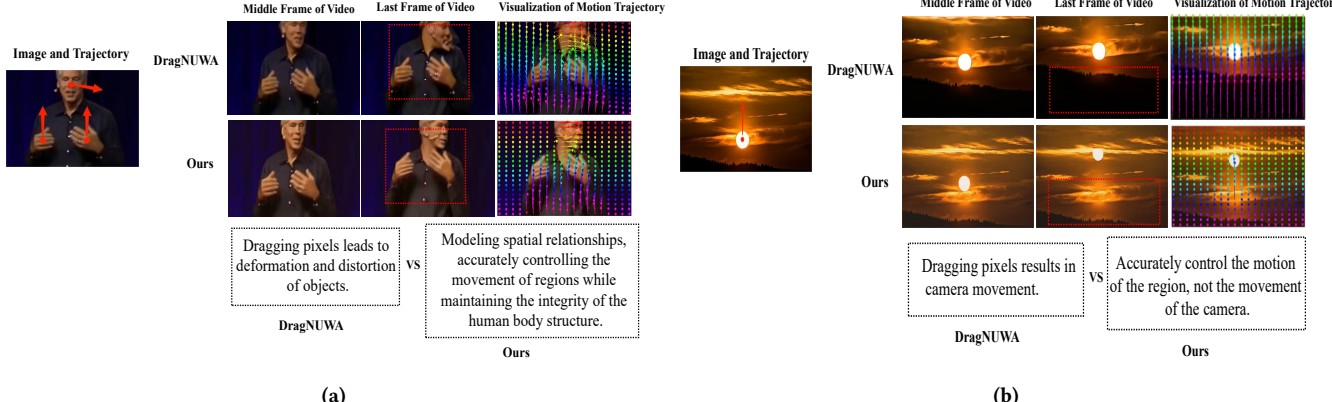

**(a)**          **(b)**

**Figure 2: Experiments on the motivation for entity representation. Existing methods (DragNUWA and MotionCtrl ) involve directly dragging pixels, which cannot precisely control the target, leading to camera motion or target structure distortion. In contrast, our method utilizes entity representation and models spatial relative positions to achieve accurate control.**

## 2.2 Controllable Video Generation

Early trajectory-based works [3, 4]often utilized optical flow or recurrent neural networks to achieve motion control. TrailBlazer [20] focuses on enhancing controllability in video synthesis by employing bounding boxes to guide the motion of subject. DragNUWA [41] encodes sparse strokes into a dense flow space, subsequently employing this as a guidance signal to control the motion of objects. Similarly, MotionCtrl [36] directly encodes the trajectory coordinates of each object into a vector map, using it as a condition to control the object's motion. These works can be categorized into two paradigms: Trajectory Map (point) and box representation. The box representation (e.g., TrailBlazer[20]) only handle instance-level objects and cannot accommodate backgrounds such as starry skies. Existing Trajectory Map Representation (e.g., DragNUWA, MotionCtrl, DragAnything) methods are quite crude, as they do not consider the semantic aspects of entities. In other words, a single point cannot adequately represent an entity. In our paper, we introduce DragEntity, which can achieve true entity-level motion control using the proposed entity representation.

## 3 METHOD

### 3.1 Task Definition and Motivation

**Task Definition.** The task of trajectory-based controllable video generation requires the model to generate videos based on a given image and motion trajectory. Given a point trajectory $(x_1, y_1)$, $(x_2, y_2)$, ..., $(x_L, y_L)$, where $L$ represents the length of the video, the conditional denoising autoencoder $\epsilon_\theta(z, c)$ is utilized to generate videos corresponding to the motion trajectory. In this paper, the guiding signal $c$ contains two types of information: trajectory points and the first frame of the video.

**Motivation.** Recently, some trajectory control methods such as DragNUWA [41] and MotionCtrl [36] use trajectory points to control the motion of objects. These methods typically manipulate the corresponding pixels or pixel areas directly using the provided trajectory coordinates or their derivatives. However, they overlook a critical issue: as shown in Figure 1, the pixels or pixel areas directly

manipulated by the trajectory do not necessarily represent the entity we intend to control. Consequently, dragging these points does not enable object motion based on trajectory control. As illustrated in Figure 1, we visualize the trajectory changes of each pixel in the generated video based on the co-tracker [17]. We can observe that:

(1) It is evident that a single pixel or a group of pixels on an object cannot fully represent the entity (Figure 3 (b)). From the pixel motion trajectory k of DragNUWA, it is clear that dragging a pixel of the sun does not lead to the movement of the sun; instead, it results in the camera moving upwards. This adequately demonstrates that a single pixel or a few pixels cannot represent the entire sun, and therefore, the model cannot understand the true meaning of our trajectory. We adopt an entity representation of objects as a more direct and effective way to precisely control the area we manipulate (the selected area), while the rest of the image remains unchanged.

(2) As shown in Figure 1(a), when multiple trajectories act on the same object, the motion of each part must maintain relative spatial relationships to preserve the object's structural integrity. By comparison, we observed that in videos synthesized by DragNUWA, different parts of the human body move independently under the control of trajectories, resulting in abnormal distortions. However, what we expect is for the human body to perform various combined bodily movements as a whole under the guidance of multiple trajectories, rather than each body part acting separately.

Based on these new insights and observations, we propose a new entity representation that integrates the spatial relationships of objects, extracting the latent features of the objects we want to control for their representation. As shown in Figure 3, the visualization of motion trajectories indicates that our method can achieve more precise motion control. For example, in Figure 3(a), our method can accurately control the combined movements of turning the head and raising the hand, whereas DragNUWA only drags the corresponding pixel areas for motion, without considering the relationships between different parts, leading to abnormal appearance distortions. In Figure 3(b), our method can accurately

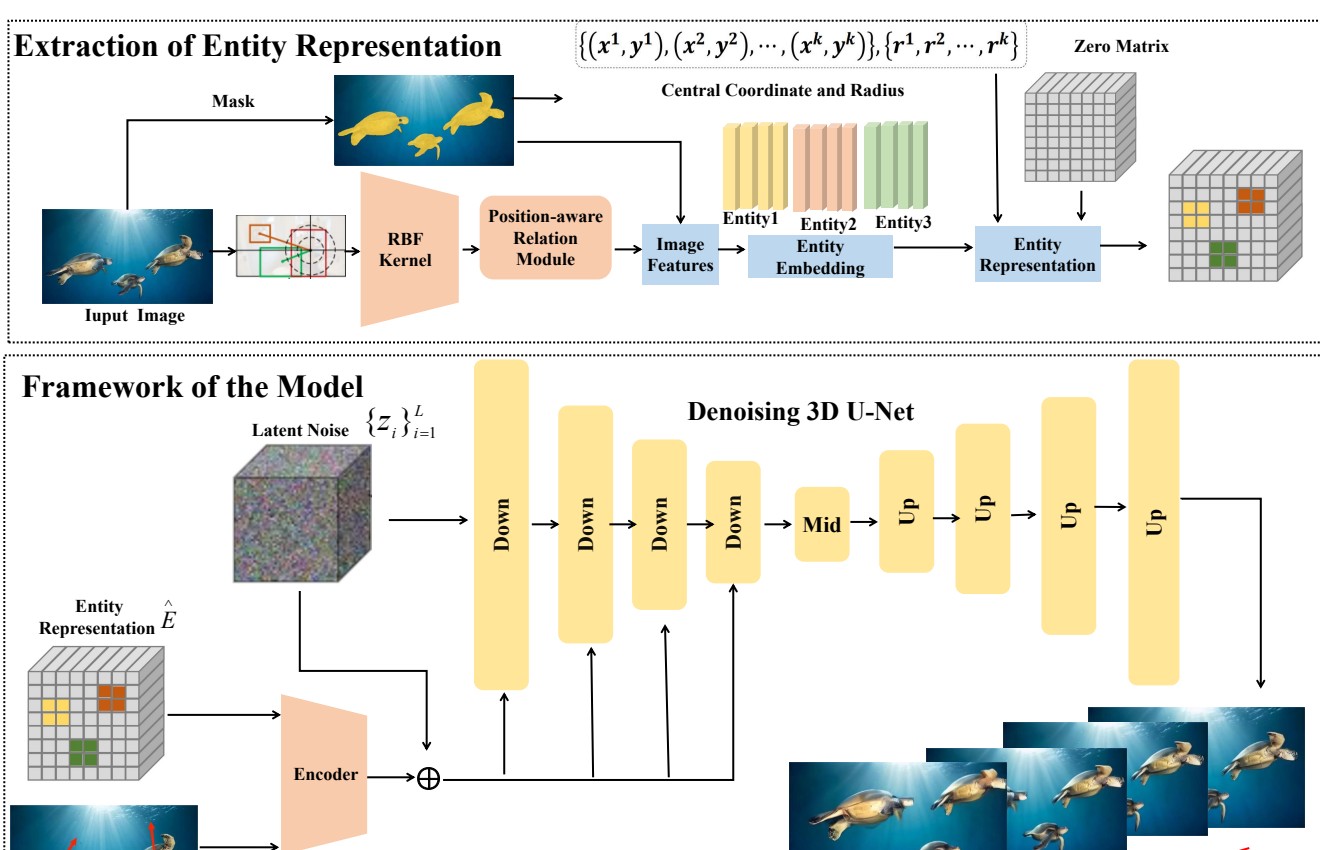

**Figure 3: Model Framework. This image consists of two parts: (a) Entity Semantic Representation Extraction. Latent features are extracted based on entity mask indices, integrating the relative spatial relationships between objects to form their respective entity representations. (b) Main Framework. Based on the SVD[2] model, it utilizes the corresponding entity representations to precisely control motion.**

control the rising of the sun, while DragNUWA interprets it as a camera displacement.

Based on the SVD [2] model, the architecture of the model primarily consists of three parts: a denoising diffusion model (3D U-Net[24]) that learns the denoising process in both spatial and temporal dimensions, an encoder and decoder that encode the supervisory signal into the latent space, and reconstruct the denoised latent features back into video. Inspired by ControlNet[43], we use a 3D Unet to encode our guiding signal, then apply it to the decoder block of the denoising 3D Unet of SVD, as shown in Figure 4. Unlike previous work, we have designed an entity representation mechanism that integrates the relative spatial relationships of objects, enabling trajectory-based controllable generation.

## 3.2 Entity Representation includes Spatial Relationships

The conditional signal of our method requires corresponding entity representations. In this section, we will describe how to extract these representations from the first frame of the image.

**Position-aware relation.** Inspired by ParNet[40], we propose a position-aware relationship module that simultaneously captures semantic and spatial object-level relationships. It is designed to enrich the representation of objects by adaptively focusing on the spatially relevant and semantically relevant parts of the input image.

In the spatial relationship branch, for each object $i$, a polar coordinate system is centered at $i$, and the four-dimensional bounding box position $p_j = (x, y, w, h)$ of object $j$ is transformed into a polar coordinate vector $(\rho_j, \theta_j)$. This is because representing the spatial direction between the centers of bounding boxes $i$ and $j$ is very effective. We believe that in describing the positional relationships between objects, absolute positions are rarely used. Instead, relative positions are widely utilized (for example, the head is directly above

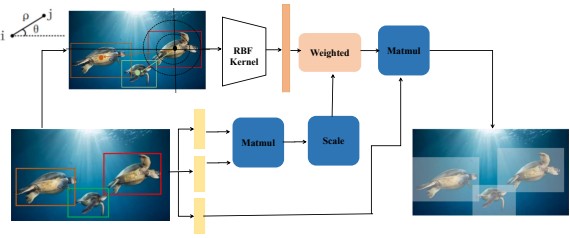

**Figure 4: Image Position-Aware Relationship Module. The entity representation includes more information about the relative spatial relationships between objects.**

both arms, etc.). Therefore, our method focuses on the relative positions of objects rather than their absolute positions.

Low-dimensional relative positions are embedded to higher dimensions through a set of Gaussian kernels with learnable means and covariances of Gaussian distributions, where the spatial relation between objects $i$ and $j$ is easily separable. The spatial relation dimension after embedding is $d_p = 64$ experimentally. The kernel operator for object $j$ centered at $i$ is defined as follows:

$$\omega_{\rho_j} = \exp\left(-\frac{\|\rho_j - \rho_0\|^2}{2\sigma_\rho^2}\right) \quad (1)$$

$$\omega_{\theta_j} = \exp\left(-\frac{\|\theta_j - \theta_0\|^2}{2\sigma_\theta^2}\right) \quad (2)$$

Where $\rho_0$ and $\sigma_\rho$ are the learnable means and covariances of Gaussian distributions for relevant distance, and $\theta_0$ and $\sigma_\theta$ are the learnable means and covariances of Gaussian distributions for relevant angle.

We aggregate the relative distance and angle relationship of objects $i$ and $j$ with a scaling function, which means that the strength of spatial relationships between objects can be weighted by spatial orientation. The aggregated spatial weight for the image is represented as $\omega_p$:

$$\omega_p = \frac{\sum_{j=1}^{N} \omega_{\rho_j} \omega_{\theta_j}}{\sum_{j=1}^{N} \omega_{\rho_j} \omega_{\theta_j}} \quad (3)$$

where $N$ is the number of objects in the image.

In another branch, image features $V$ is linearly transformed by $f(\cdot)$. Semantic relation $\omega_s$ is computed as in Eq. (4). Dot-product attention is employed in our algorithm with a scaling factor $\frac{1}{\sqrt{d_v}}$, where $d_v$ is the dimension of object features $v$.

$$\omega_s = \frac{f(V)^T f(V)}{\sqrt{d_v}} \quad (4)$$

where $d_v$ is the dimension of object feature $V$.

The intra-image relation weight $\omega_I$ indicates both the semantic and spatial impact from object $j$. Spatial relationship $\omega_p$ of different objects is fused with semantic relationship $\omega_s$ between objects through Eq. (5). It is scaled in the range (0, 1) and can be regarded as a variant of softmax. $\omega_I$ is computed as follows:

$$\omega_I = \frac{\omega_p \exp(\omega_s)}{\sum_{k=1}^{N} \omega_p \exp(\omega_s)} \quad (5)$$

Multi-head attention[31] is employed to adapt flexible relationships, since different heads can focus on different aspects of relation. Multiple relation features from multi heads are aggregated as follows:

$$V_r = f(V) + \text{Concat}[(\omega_I f(V))_1, (\omega_I f(V))_2, \ldots, (\omega_I f(V))_K] \quad (6)$$

$K$ is the number of relation heads, which is typically set to be 6, same as transformer.

**Entity Representation.** Using the image features $V$, the corresponding entity embeddings can be obtained by indexing the coordinates from the segmentation mask. For convenience, average pooling is used to process the corresponding entity embeddings, resulting in the final embeddings $\{e_1, e_2, \ldots, e_k\}$, where $k$ represents the number of entities, and the channel size of each entity is $C$.

To associate these entity embeddings with the corresponding trajectory points, we directly initialize a zero matrix $E \in \mathbb{R}^{H \times W \times C}$ and then insert the entity embeddings based on the trajectory sequence points, as shown in Figure 5. During the training process, we use the entity mask of the first frame to extract the center coordinates $\{(x_1, y_1), (x_2, y_2), \ldots, (x_k, y_k)\}$ of the entity as the starting point for each trajectory sequence point. With these center coordinate indices, the final entity representation $\hat{E}$ can be obtained by inserting the entity embeddings into the corresponding zero matrix $E$.

With the center coordinates $\{(x_1, y_1), (x_2, y_2), \ldots, (x_k, y_k)\}$ of the entity in the first frame, we use Co-Tracker to track these points and obtain the corresponding motion trajectories $\{(x_{ki}, y_{ik})\}_{i=1}^{L}$, where $L$ is the length of the video. Then we can obtain the corresponding entity representation $\{\hat{E}_i\}_{i=1}^{L}$ for each frame.

**Encoder for Entity Representation.** In this encoder, we utilized four blocks of convolution to process the corresponding input guidance signal, where each block consists of two convolutional layers and one SiLU activation function. Each block downsamples the input feature resolution by a factor of 2, resulting in a final output resolution of $\frac{1}{8}$. The encoder structure for processing the entity representation is the same, with the only difference being the number of channels in the first block, which varies when the channels for the two representations are different. After passing through the encoder, we follow ControlNet [52] by adding the latent features of Entity Representation Map Representation with the corresponding latent noise of the video:

$$\{R_i\}_{i=1}^{L} = \text{encoder}(\{\hat{E}_i\}_{i=1}^{L}) + \{Z_i\}_{i=1}^{L}, \quad (7)$$

where $Z_i$ denotes the latent noise of the $i$-th frame. Then the feature $\{R_i\}_{i=1}^{L}$ is inputted into the encoder of the denoising 3D Unet to obtain four features with different resolutions, which serve as latent condition signals. The four features are added to the feature of the denoising 3D Unet of the foundation model.

## 3.3 Training and Inference

During the training process, we need to generate corresponding Trajectories of Entity Representation and 2D Gaussian, as shown in Figure 5. First, for each entity, we calculate its incircle circle using its corresponding mask, obtaining its center coordinates $(x, y)$ and radius $r$. Then we use Co-Tracker to obtain its corresponding trajectory of the center $\{(x_i, y_i)\}_{i=1}^{L}$, serving as the representative motion trajectory of that entity. With these trajectory points and radius,

we can calculate the corresponding Gaussian distribution value at each frame. For entity representation, we insert the corresponding entity embedding into the circle centered at $(x, y)$ coordinates with a radius of $r$. Finally, we obtain the corresponding trajectories of Entity Representation and 2D Gaussian for training our model.

**Loss Function.** In video generation tasks, Mean Squared Error (MSE) is commonly used to optimize the model. Given the corresponding entity representation $\hat{E}$ and 2D Gaussian representation $G$, the objective can be simplified to:

$$\mathcal{L}_\theta = \sum_{i=1}^{L} \mathbf{M} \left\| \epsilon - \epsilon_\theta \left( \boldsymbol{x}_{t,i}, \text{cond}(\hat{\mathbf{E}}_i), \text{cond}(\mathbf{G}_i) \right) \right\|_2^2, \quad (8)$$

where $E_\theta$ denotes the encoder for entity and 2D Gaussian representations. $M$ is the mask for entities of images at each frame. The optimization objective of the model is to control the motion of the target object. For other objects or the background, we do not want to affect the generation quality. Therefore, we use a mask $M$ to constrain the MSE loss to only backpropagate through the areas we want to optimize.

**Inference of User-Trajectory Interaction.** During the inference process, users simply need to click to select the area they want to control with SAM [18] . For the human body, this is further refined into segmented areas for each part. Then, by dragging any pixel within that area, they form a reasonable trajectory. Our model can then accurately control the area based on this trajectory to generate a video corresponding to the desired motion.

## 4 EXPERIMENTS

### 4.1 Experiment Settings

**Implementation Details.** All our training is based on the stable Video Diffusion (SVD) [2] architecture and weights, which have been trained to generate 20 frames at a resolution of 320×576. All experiments were conducted on PyTorch, using Tesla A100 GPUs. AdamW was used as the optimizer for a total of 100k training steps, with a learning rate of 1e-6.

**Evaluation Metrics.** We adopt two types of evaluation metrics: 1) Video Quality Assessment: We use the Frechet Inception Distance (FID) and Frechet Video Distance (FVD) [30]to evaluate the visual quality and temporal coherence. 2) Object Motion Control Performance Evaluation: The Euclidean distance between the predicted object trajectory and the ground truth trajectory (ObjMC) is used to evaluate the object motion control. Furthermore, in user studies, we randomly generated 50 videos and had 10 non-professional individuals vote on the quality of the generated videos and the motion trajectories.

**Datasets.** We adopt VIPSeg[21], WebVid[22] and TED-talks[26] as our testsets. Since FVD requires videos to have the same resolution and duration, we resized the VIPSeg validation dataset to 256×256 and reduced its length to 14 frames for assessment. We use the VIPSeg and WebVid as our training sets, and employed a collaborative tracker to obtain the corresponding motion trajectories, which served as annotations.

## 4.2 Comparative Experiments

**Comparisons with State-of-the-Art Methods.** Video Quality Evaluation on on the WebVid datasets. Table 1 presents a comparison of video quality on the WebVid datasets using the FID metric. We controlled for other conditions being the same (basic architecture) and compared the performance between our method and DragNUWA. Our FID score reached 34.5, significantly outperforming the current SOTA model DragNUWA (34.5 vs. 36.9). According to MotionCtrl[36], motion control performance is evaluated using ObjMC by calculating the Euclidean distance between predicted trajectories and ground truth trajectories. Compared to DragNUWA, our model achieved state-of-the-art performance with a score of 302.7. Additionally, FVD assesses the temporal coherence of generated videos by comparing the feature distribution between generated videos and ground truth videos. Compared to the performance of DragNUWA (521.7), our model also achieved superior temporal coherence with a score of 510.8, significantly improving by 10.9.

At the same time, we also conducted visual comparisons on the TedTalk and VIPSeg datasets. As shown in Figure 5, we can observe that when DragNVWA uses multiple trajectories to control the motion of characters, it results in distortion (third and seventh rows), artifacts (fifth row), and multiple hands (first row). As shown in Figure 6, when DragNVWA controls multiple objects, there are issues such as appearance distortion (third row), incorrect movement directions (first row), and incorrect camera movements (fifth row), but our model DragEntity can precisely control motion.

**Table 1: Performance Comparison on the WebVid Dataset**

| Method | ObjMC↓ | FVD↓ | FID↓ |
|---|---|---|---|
| MotionCtrl | 350.6 | 584.2 | 41.8 |
| DragNUWA | 326.5 | 521.7 | 36.9 |
| Ours | **302.7** | **510.8** | **34.5** |

**Ablation Studies.** Entity representation and Spatial Relationship are core components of our work. We keep other conditions constant and only modify the corresponding condition embedding features. Table 2 shows the ablation studies of these two aspects. To study the influence of entity representation, we observe performance changes by determining whether this representation is included in the final embedding. Since the entity representation primarily affects the object motion in generated videos, we only need to compare ObjMC, while FVD and FID metrics focus on temporal consistency and overall video quality. With entity representation, the model's ObjMC saw significant improvement, reaching 311.4.

**Table 2: Ablation Study for Entity representation and Spatial Relationship.**

| Entity Rep. | Position | ObjMC↓ | FVD↓ | FID↓ |
|---|---|---|---|---|
|  |  | 368.7 | 563.3 | 42.2 |
| ✓ |  | 311.4 | 527.5 | 36.1 |
|  | ✓ | 339.3 | 538.3 | 40.9 |
| ✓ | ✓ | 302.7 | 510.8 | 34.5 |

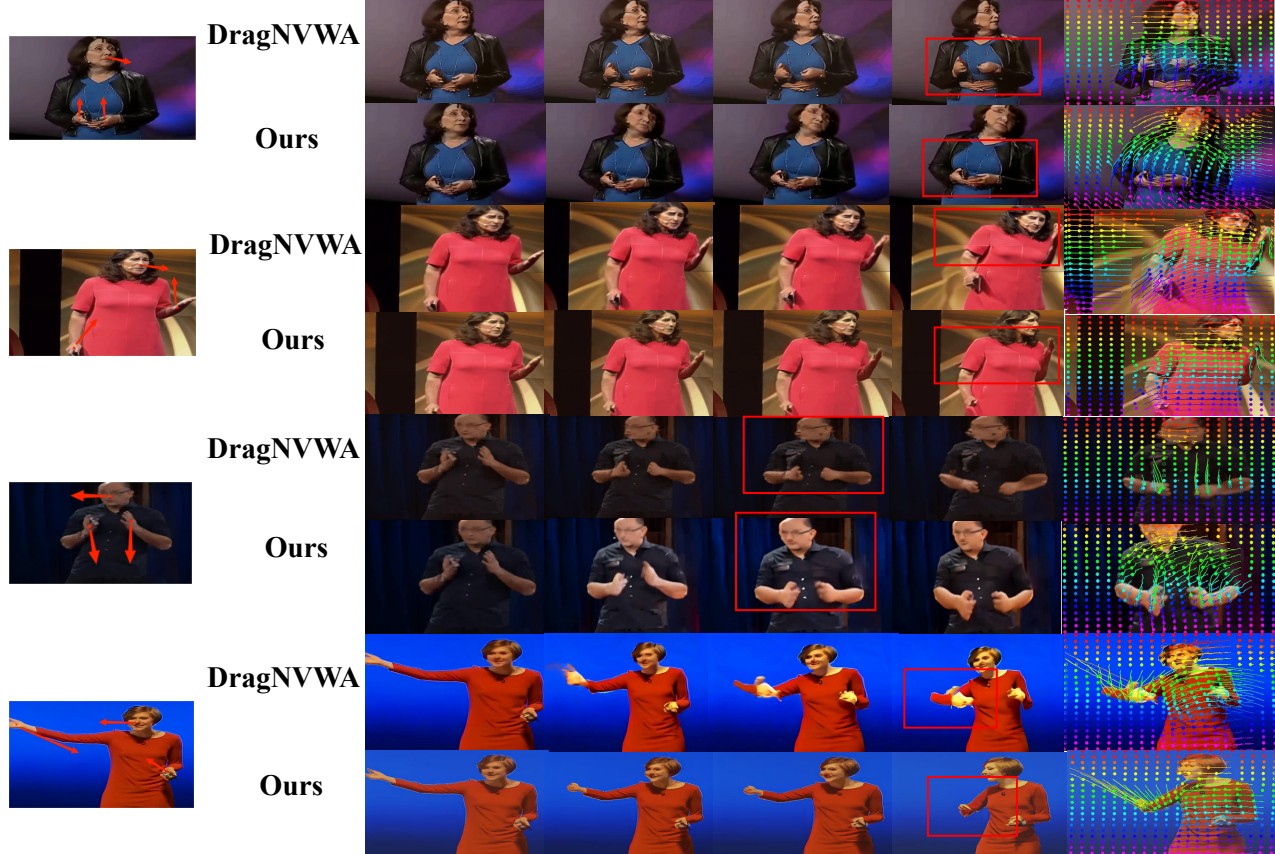

**Figure 5: Visual comparison on the TedTalk dataset. It can be observed that when multiple trajectories are active simultaneously on the human body, the DragNVWA model exhibits phenomena such as character distortion (third and seventh rows), artifacts (fifth row), and multiple hands (first row). Our model, while maintaining the basic skeleton of the human body, is able to move accurately according to the trajectories.**

Similar to entity representation, we observe changes in ObjMC performance by determining whether the final embedding includes spatial position relationships. The spatial relationships between objects led to a performance improvement of 29.4, reaching 339.3. Overall, the highest performance occurs when both entity representation and spatial position relationships are used together, achieving 302.7. This phenomenon indicates that these two representations have a mutually reinforcing effect, contributing to the precise control of the trajectory.

Additionally, we also explored the impact of the loss mask. Table 3 presents the ablation study for Loss Mask. When the loss mask is not used, we directly optimize the MSE loss for each pixel of the entire image. The loss mask can bring certain benefits, approximately improving ObjMC by 13.4.

**User Evaluation** We conducted a user survey to assess video authenticity from MotionCtrl, DragNVWA, and our method, each at 576×320 resolution. Ten volunteers chose the best method based on: 1) structural integrity; 2) trajectory consistency; 3) overall quality.

**Table 3: Ablation Study for Loss Mask. Loss mask can bring certain gains, especially for the ObjMC metric.**

| Loss Mask $M$ | ObjMC↓ | FVD↓ | FID↓ |
|---|---|---|---|
|  | 316.1 | 516.2 | 36.3 |
| ✓ | 302.7 | 510.8 | 34.5 |

Table. 4 shows our method outperforms others in detail, motion, and quality.

## 5 CONCLUSION

In this paper, we introduce a new trajectory-based motion control method and present two new insights: 1) Pixel points controlled by trajectories do not adequately represent entities. 2) When multiple trajectories act on the same object, the motion of each part must maintain relative spatial relationships to preserve the object's structural integrity. To address these two technical challenges, we

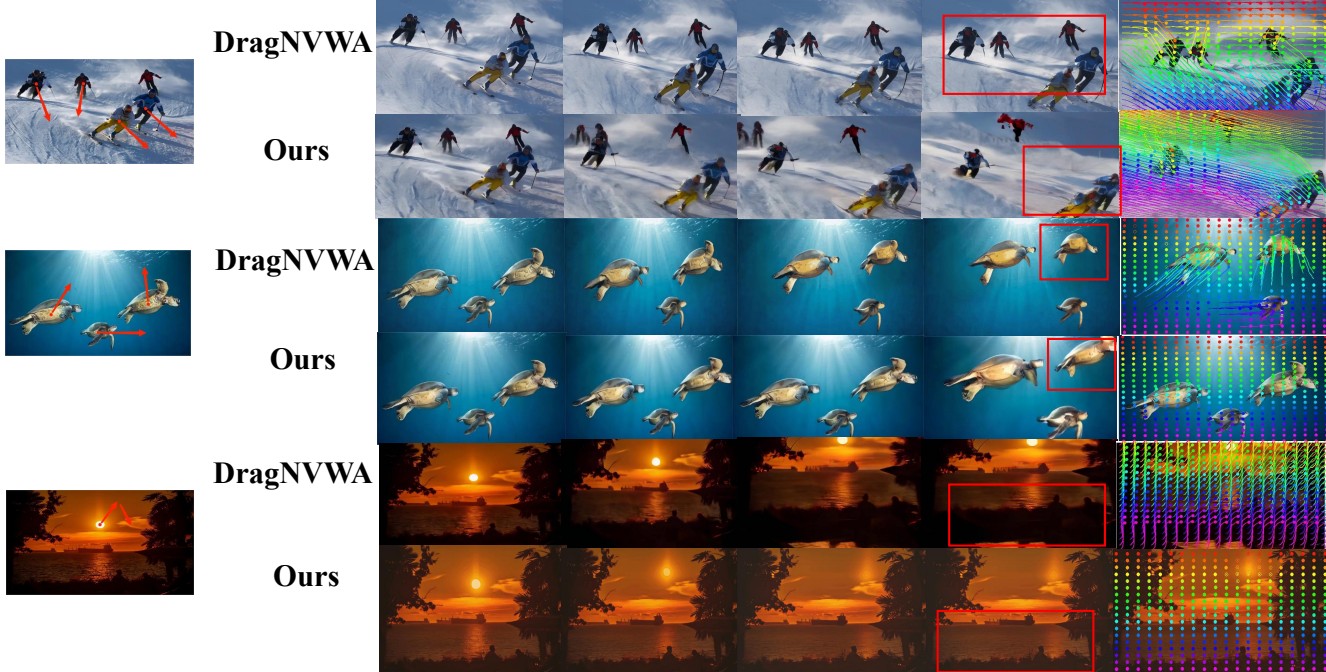

**Figure 6: Visual comparison with DragNUWA. DragNUWA results in appearance distortion (third row), incorrect movement direction (first row), and incorrect camera movement (fifth row), whereas DragEntity can precisely control movement.**

**Table 4: User study results. The percentages indicate the proportion of 10 volunteers who selected the best result from five methods, evaluated from three different perspectives.**

| Evaluation Criteria | MotionCtrl | DragNVWA | Ours |
|---|---|---|---|
| Structural integrity | 10% | 25% | **65%** |
| Trajectory consistency | 20% | 35% | **45%** |
| Overall feeling | 10% | 30% | **60%** |

propose DragEntity, which utilizes latent features to represent each entity. Our proposed entity representation, which incorporates spatial relative position relationships as a self-domain embedding, can control the motion of entities in the image while maintaining structural integrity. Experiments validate the superiority of our method over existing approaches, demonstrating its ability to effectively generate fine-grained videos.

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
