# OpenReview forum: "DragEntity:Trajectory Guided Video Generation using Entity and Positional Relationships"
_acmmm.org/ACMMM/2024/Conference — MM2024 Oral_

### Official Review · Reviewer_AxYa · 2024-05-16

**Rating:** 5
**Confidence:** 3

**Summary:**

The paper proposes to improve the trajectory guided video generation by using entity representations, rather than point/pixel/bounding box representations. Specifically, the paper utilizes stable video diffusion and CoTrack to segment the frames to identify some entities and track their motion trajectories, which allows users to drag them to control the video generation. Meanwhile, the proposed method also employs a position-aware relation module to model and constrain the relative positions or motion among multiple object parts or entities. The proposed method has been evaluated on 3 public datasets and compare favorably with the SOTA.

**Strengths:**

The paper is well motivated since how to control the video generation by the motion of multiple objects is a critical problem for users to adopt the video generation technology.

The proposed entity representation and position-aware relation module are new and sound approaches to advance the trajectory guided video generation.

The experiments are convincing to demonstrate the clear advantage of the proposed method.

Overall, this is a descent paper which is a clear step to advance  trajectory guided video generation.

**Limitations:**

I wonder how sensitive of the RBF kernel and the learnable means and covariances, as well as the semantic relation in Eq.4 to the video generation performance. Even for the same kind of objects, the motion or relative positions among the parts could be dramatically different due to the camera view angles. Please elaborate on the generalizability of the position-aware relation, since the training videos may not cover all possible motion and view angles of objects.

Typo: "on on" line 641.

**Suitability:**

3

---

### Official Review · Reviewer_FG3d · 2024-05-24

**Rating:** 4
**Confidence:** 3

**Summary:**

This paper tackles controllable video generation via entity representation to control the motion of multiple objects. To ensure more user-friendly interaction, the proposed DragEntity method utilizes trajectories to control objects in the image. The method is compared with previous work DragNUWA. The results on object moving shows that DragEntity is able to perform fine-grained control with better performance compared with DragNUMA and MotionCtrl in terms of several evaluation metrics. However, additional clarification along with experimental comparisons may be needed for better evaluations of the proposed video generation method.

**Strengths:**

1. DragEntity focuses on trajectory-based control of video generation, which has emerged to be user-friendly regarding content guidance for movement. By leveraging the latent features to represent individual entities, DragEntity considers entity-level motion and relative spatial positions between objects to ensure the structural and motion integrity of the moving object.
2. Multiple evaluation metrics are involved to evaluate the quality of generated videos from different perspectives.

**Limitations:**

1. In Section 4.1, the “Datasets” paragraph mentions that “VIPSeg[21], WebVid[22] and TED-talks[26]” are utilized as test sets, but only quantitative results on WebVid are given as in Table 1. What about the results on the other two datasets?
2. Figure 6, the sun disappears in the last example for the proposed method. Since this example focuses on the movement of the sun, how does the disappearance of the moving objects demonstrate the performance of DragEntity?
3. Could DragEntity be compared with methods that are able to perform image editing that includes object moving, e.g., DragGAN and DragDiffusion?
4. The first sentence of the paragraph above the itemized contributions in the introduction mentions SVD [2] as “the base segmentation model”. This description may require a clarification since SVD is known as Stable Video Diffusion model for video generation, not for segmentation. For the extraction of entity representation, how the mask of each entity is acquired? If the DragEntity method assumes a given mask, it may be unfair to compare this with existing video generation methods.

**Suitability:**

3

---

### Official Review · Reviewer_FMkh · 2024-06-04

**Rating:** 4
**Confidence:** 3

**Summary:**

This paper proposes a method for trajectory guided video generation with a new entity representation. It first extracts the entity representation with a proposed module, then the proposed main framework will utilize the entity representations to precisely control motion in the generated video based on a previous model SVD.

**Strengths:**

The entity representation designed to enrich the representation of objects by adaptively focusing on the spatially relevant and semantically relevant parts of the input image is novel.

With the new entity representation, authors show this method can perform better on issues like deformation and distortion of objects, camera movement, etc.

**Limitations:**

Authors show several cases that the proposed method can perform better than other methods:

For the camera movement issue, the comparison with DragNVWA may not be fair. If I understand correctly, proposed method is based on SVD segmentation, which already implicitly contains rich segmentation information beyond the training data of DragNVWA, while DragNVWA is just using optical flow within the training data. DragNUWA’s camera movement problem in Fig. 2 may because there’s few sun rise videos in the training dataset, mostly camera movement. But your method is able to do it may because the SVD can segment the sun. Is the proposed entity representation with spatial relationships play a key role here? What if DragNVWA + SVD segmentation? Ablation study may be needed.

For the object deformation/distortion issue, I’m not convinced why the designed method can resolve this issue. Like Fig. 2, the deformation/distortion is just in the head area, maybe proposed method can segment the head with the LIP, but how can the relationships between objects/components help resolve deformation/distortion within one object/component?

The main issue of this paper to me is, the reason why the proposed method can solve the issues is not fully explained with descriptions/experiments.

Another issue is, the terminology is not consistent, which may lead to confusion. For example, in Sec. 3.2 it says “Entity Representation includes Spatial Relationships”, but in the ablation study, it says “Entity representation and Spatial Relationship are core components of our work”, I’m not sure how you do the experiment only with spatial relationship.

Also, the experiments may not be enough. Only table 1 is the comparison table with other method on 1 dataset, more comparison will be more convincing.


Other minor issues/comments:

In abstract, MotionCtrl in line 17 should be DragEntity?

Fig 6, last image of the 4th row, should show the optical flow for your method, here seems you are using the original one?

Fig 6, first and second row comparison, DragNVWA’s movement direction looks ok to me (maybe not as much as your method), and DragNVWA actually has less distortion (number of human, shape of human), so I would say DragNVWA looks better in this case.

**Suitability:**

3

---

### Official Review · Reviewer_owyx · 2024-06-04

**Rating:** 4
**Confidence:** 2

**Summary:**

This article introduces a video generation model called DragEntity, which uses entity representation to control the motion of multiple objects in a video. Compared to previous methods, DragEntity offers a user-friendly interaction method and can simultaneously and accurately control multiple objects moving along complex trajectories while maintaining their relative spatial relationships. Experiments have verified its effectiveness and superior performance in fine-grained control for video generation.

**Strengths:**

1. DragEntity allows users to interact with the video generation model by drawing trajectories, simplifying the user operation process.
2. Unlike pixel-based methods, DragEntity achieves true entity-level motion control, ensuring the structural integrity of objects during the dragging process.
3. Unlike pixel-based methods, DragEntity achieves true entity-level motion control, ensuring the structural integrity of objects during the dragging process.
4. DragEntity introduces modeling of relative spatial relationships between objects, preventing the generation of highly unrealistic motion videos caused by trajectory dragging.
5. The model can accurately control the motion of regions rather than camera motion, avoiding deformation or distortion of objects that can result from direct pixel dragging.

**Limitations:**

1. Judging from the results, the method's dragging of human bodies appears to be quite eerie. The quality of generation does not seem to be significantly better. Has the author provided an explanation for this issue, as it does not substantiate the contribution points proposed in the paper?
2. The discussion of the method in the paper seems rather limited in the experiments. Has the author further explored the model's generalization ability, computational cost, and the finesse of control?

**Suitability:**

3

---

### Meta-Review · Area_Chair_QDCT · 2024-07-05

**Recommendation:** Accept (Oral)
**Confidence:** 4

**Metareview:**

The authors introduces DragEntity, a user friendly interaction method,  to control the motion of multiple objects in a video using entity representation while preserving spatial relationships. The authors demonstrate the superior performance through experiments.

The reviewers identify several strengths of the proposed method along with some weaknesses (see details in the reviews). The authors agree on the suitability of the paper for the ACM MM conference. While some concerns from the reviewers remain after the rebuttal, the quality of the paper is suitable enough to be accepted provided some of the limitations are addressed by the camera ready version.